Integrative taxonomy reveals three new species and one new record of Psychropotes (Holothuroidea, Elasipodida, Psychropotidae) from the Kermadec Trench region and the Wallaby-Zenith Fracture Zone

Xiao Yunlu 1 2
Zhang Haibin 1 3 hzhang@idsse.ac.cn
1 Institute of Deep-sea Science and Engineering, Chinese Academy of Sciences , Sanya , China
2 University of Chinese Academy of Sciences , Beijing , China
3 HKUST-CAS Sanya Joint Laboratory of Marine Science Research, Chinese Academy of Sciences , Sanya , China
Woo Sau Pinn
Electronic publication date: 2025 Feb 21
Publication date: 2025
Volume: 13
Electronic Location ID: e18806
Received 2024 Sep 6; Accepted 2024 Dec 12
Copyright: © 2025 Xiao and Zhang
Copyright year: 2025
Copyright holder: Xiao and Zhang
License: This is an open access article distributed under the terms of the Creative Commons Attribution License, which permits unrestricted use, distribution, reproduction and adaptation in any medium and for any purpose provided that it is properly attributed. For attribution, the original author(s), title, publication source (PeerJ) and either DOI or URL of the article must be cited.
License URL: https://creativecommons.org/licenses/by/4.0/

Keywords: Deep-sea, Hadal zone, Morphology, New species, Phylogeny, Systematics

Funding: Major scientific and technological projects of Hainan Province ZDKJ2021036 National Key Research and Development Program of China 2023YFC2809300, 2022YFC2805400 International Collaboration Program of CAS 183446KYSB20210002 National Institute of Water & Atmospheric Research (NIWA), New Zealand IDS23301 This study was supported by the Major scientific and technological projects of Hainan Province (ZDKJ2021036), the National Key Research and Development Program of China (2023YFC2809300, 2022YFC2805400), the International Collaboration Program of CAS (grant nos. 183446KYSB20210002), and the National Institute of Water & Atmospheric Research (NIWA), New Zealand (project IDS23301). The funders had no role in study design, data collection and analysis, decision to publish, or preparation of the manuscript.

==============================
The holothuroid genus Psychropotes is the largest genus in the family Psychropotidae. Prior to this study, this genus contained 20 accepted species and was prominent representatives in the deep-sea benthic fauna at lower bathyal-abyssal depths throughout the global oceans, but it has been poorly studied in the hadal zone. Deep-sea holothuroids were collected in October 2022 to March 2023 by the joint China-New Zealand deep-diving scientific expedition to the Kermadec Trench in the South Pacific Ocean and the Wallaby-Zenith Fracture Zone in the East Indian Ocean at a depth of 6,018–6,605 m. Our examination of specimens of Psychropotes revealed three new species, which we described as Psychropotes diutiuscauda sp. nov., Psychropotes nigrimargaria sp. nov., and Psychropotes asperatus sp. nov. We also recorded Psychropotes depressa (Théel, 1882) for the first time from the Kermadec Arc at a depth of 1,620 m. We provide comprehensive descriptions of the morphological features and a taxonomic key for the genus Psychropotes. We also performed a phylogenetic analysis of the order Elasipodida based on COI sequences and the concatenated 16S-COI sequences. Intraspecific and interspecific genetic distances were calculated among Psychropotes Species based on COI sequences. Our phylogenetic analyses supported the assignment of three new species to the genus Psychropotes and their differentiation from congeners. The geographical distribution and global depth range of psychropotid species were summarized, and the results showed that the Pacific region had the highest species diversity. These findings contribute to the taxonomic diversity and patterns of geographical distribution in the family Psychropotidae.

Introduction

Holothuroids, also known as sea cucumbers, are the third most diverse class among echinoderms, with seven ordinal rank clades (Miller et al., 2017). They are primitive benthic animals distributed at all depths from shallow water to hadal zones (Iken et al., 2001; Jamieson et al., 2011; Lee et al., 2017), and they are abundant in many marine benthic ecosystems, especially in the hadal zone (Kuhnz et al., 2014). Among the seven orders in the class Holothuroidea, the order Elasipodida Théel, 1882, which includes four families Psychropotidae Théel, 1882; Elpidiidae Théel, 1882; Laetmogonidae Ekman, 1926; and Pelagothuriidae Ludwig, 1893, is unique in being confined to the deep sea (Hansen, 1975). The family Psychropotidae includes 42 species (e.g., Koehler & Vaney, 1905; Hansen, 1975; Rogacheva, Cross & Billett, 2009; Gebruk, Kremenetskaia & Rouse, 2020; Yuan, Wang & Zhang, 2024), which belong to three genera (Psychropotes Théel, 1882; Benthodytes Théel, 1882; and Psycheotrephes Théel, 1882). As the largest genus in Psychropotidae, Psychropotes contains 20 valid species (WoRMS, 2024): Psychropotes belyaevi Hansen, 1975; P. buglossa Perrier, 1886; P. depressa (Théel, 1882); P. dubiosa Ludwig, 1893; P. dyscrita (Clark, 1920); P. fuscopurpurea Théel, 1882; P. hyalinus Pawson, 1985; P. longicauda Théel, 1882; P. loveni Théel, 1882; P. minuta Koehler & Vaney, 1905; P. mirabilis Hansen, 1975; P. monstrosa Théel, 1882; P. moskalevi Gebruk & Kremenetskaia in Gebruk, Kremenetskaia & Rouse, 2020; P. pawsoni Gebruk & Kremenetskaia in Gebruk, Kremenetskaia & Rouse, 2020; P. raripes Ludwig, 1893; P. scotiae (Vaney, 1908); P. semperiana Théel, 1882; P. verrucicaudatus Xiao et al., 2019; P. verrucosa (Ludwig, 1893); and P. xenochromata Rogacheva & Billett in Rogacheva, Cross & Billett, 2009. This genus was established by Théel (1882), who erected four species discovered during the H.M.S Challenger Expedition. Species in Psychropotes are characterized by having a ventral anus, the presence of unpaired dorsal appendages, and the absence of circumoral papillae. As the distinctive members of deep-sea benthic communities at lower bathyal-abyssal depths throughout the world’s oceans (Hansen, 1975; Rogacheva, Cross & Billett, 2009; Gebruk, Kremenetskaia & Rouse, 2020), this genus has revealed new species consistently in recent years (Xiao, Li & Sha, 2018; Xiao et al., 2019; Gebruk, Kremenetskaia & Rouse, 2020; Yu et al., 2021, 2022; Xiao, Xiao & Zeng, 2023; Yuan, Wang & Zhang, 2024). This genus has also been found at hadal depths (6,135–7,290 m) across the Pacific Ocean: P. moskalevi and P. pawsoni in the northwest Pacific, and P. verrucosa in the Banda Trench (Ludwig, 1893; Hansen, 1975; Gebruk, Kremenetskaia & Rouse, 2020).

The Kermadec Trench in the South Pacific Ocean is about 1,195 km long and 120 km wide, with a maximum depth of 10,106 m (Jamieson et al., 2020; Jamieson, Giles & Stewart, 2024). Holothuroids are one of the most species-rich groups in the Kermadec Trench at hadal depths (Jamieson et al., 2020), but there are few reports and studies on deep-sea holothuroids from the trench and its region. The published records of deep-sea holothuroids from the Kermadec Trench are largely derived from the Galathea Report: Scientific results of the Danish Deep-Sea Expedition Round the World 1950–52 (Hansen, 1975). Notable contributions to our understanding of holothuroids in the Kermadec Trench region are as follows: (1) Benham (1912) provided the only report of shallow holothuroid echinoderms for the Kermadec Islands; (2) Keable & Reid (2015) reviewed the marine invertebrates collected during the Kermadec Biodiscovery Expedition 2011 and provided an inventory of invertebrates in the intertidal and subtidal habitats of the Kermadec Islands, which only included three shallow holothuroid species recorded by O’Loughlin & Vandenspiegel (2012). Five species of Psychropotes have so far been reported from the South Pacific: Psychropotes depressa, P. longicauda, P. loveni, P. monstrosa, and P. verrucosa. Three of these were reported in the Kermadec Trench: P. verrucosa, P. longicauda, and P. loveni (Hansen, 1975; Gebruk, Kremenetskaia & Rouse, 2020).

The Indian Ocean is the third largest ocean in the world, and it occupies nearly 20% of the global ocean volume, with an average depth of 3,741 m (Eakins & Sharman, 2010). There are relatively few deep-sea biological studies in the Indian Ocean, and assessment of the deep-sea benthic fauna is rare from depths >5,000 m (Parulekar et al., 1982; Janßen, Treude & Witte, 2000; Pavithran et al., 2009; Weston et al., 2021). Compared with the central and western Indian Oceans, few studies have been conducted on benthic organisms at depths >3,000 m in the eastern Indian Ocean (Post et al., 2021; Weston et al., 2021; Jamieson et al., 2022), where studies have focused mainly on canyons, the Java Trench, and Wallaby-Zenith Fracture Zone (WZFZ). The WZFZ is located in the Wharton Basin of the East Indian Ocean, with complex geomorphological features and a maximum depth of 6,625 m. In the survey of the WZFZ by Sonne in 2017, some species were reported for the first time in this area (Weston, Peart & Jamieson, 2020; Weston et al., 2021), but the taxonomic studies of the WZFZ were very limited.

From October 2022 to March 2023, a joint China-New Zealand scientific expedition carried out a large-scale and systematic manned, deep-diving investigation. During marine scientific expeditions around Oceania, researchers successfully collected more than 110 holothuroid specimens from the Kermadec Trench, its surrounding regions, and the WZFZ in the eastern Indian Ocean. This was accomplished using the Chinese-manufactured submersible ‘Fendouzhe’, deployed during Cruise TS29 aboard the research vessel research vessel Tansuo-1 (Explorer-1) (Peng et al., 2023; Zhou & Peng, 2023). Based on morphological and molecular phylogenetic analyses, three new species (i.e., Psychropotes asperatus sp. nov., Psychropotes diutiuscauda sp. nov., and Psychropotes nigrimargaria sp. nov.) and one new record (i.e., Psychropotes depressa from the Kermadec Trench region) of the genus Psychropotes were discovered. We provide detailed descriptions of these specimens, along with a taxonomic key for the genus Psychropotes. We analyzed their phylogenetic relationships within elasipodid species and the inter- and intraspecific divergence among ten species of Psychropotes. In addition, we also discuss the diversity and distribution of psychropotid holothuroids.

Materials and Methods

Sampling and preservation

The human-occupied vehicle (HOV) ‘Fendouzhe’ was used to collect Psychropotes four specimens from the Kermadec Trench region in the South Pacific and the WZFZ in the East Indian Ocean (October 2022–March 2023) (Fig. 1). Specimens were sampled and photographed in situ using HOV cameras. Sampling in the Wallaby-Zenith Fracture Zone, located in international waters, did not require a specific permit. However, sampling in the Kermadec Trench region was conducted under the appropriate permits with specimens collected under the authorization of the Ministry for Primary Industries, Special Permit 842. For morphological analyses, specimens were photographed promptly on board after collection and then fixed in 99% high grade absolute ethanol. Two specimens (i.e., IDSSE-EEB-HS175 & IDSSE-EEB-HS176) collected from WZFZ were deposited in the Institute of Deep-sea Science and Engineering (IDSSE), Chinese Academy of Sciences (CAS), Sanya, China. The other two specimens (i.e., NIWA164003 & NIWA164160) that were collected from Kermadec Trench were registered onboard using the Specify niwainvert database of National Institute of Water and Atmospheric Research (NIWA) and were loaned by the NIWA Invertebrate Collection (NIC) to IDSSE.

Figure 1 Sampling sites of the studied Psychropotes species in the Kermadec Trench and the Wallaby-Zenith Fracture Zone.

Red circle means P. diutiuscauda sp. nov.; black dot means P. nigrimargaria sp. nov.; red hexagon means P. asperatus sp. nov.; red triangle means P. depressa (Théel, 1882)

Morphological observations

General morphology was studied by means of a dissecting stereomicroscope (OLYMPUS SZX7). Ossicles were obtained from the dorsal and ventral body walls, tentacles, dorsal papillae, and tube feet by digestion of tissues in a solution of 15% sodium hypochlorite, and then they were washed with distilled water and 75% ethanol. To investigate the ultrastructure of ossicles, ossicles were air-dried, coated with gold, and observed using a scanning electron microscope (SEM). SEM imaging was performed using a Phenom ProX scanning electron microscope. For the identity and terminology of the ossicles, we followed Hansen (1975) and Gebruk, Kremenetskaia & Rouse (2020).

DNA extraction and sequencing

Total genomic DNA was extracted from small pieces of 20–30 mg holothuroid muscle tissue of each specimen using a TIANamp Marine Animals DNA Kit (TianGen, Beijing), following the manufacturer’s instructions. The PCR amplification for 5′-end of two partial mitochondrial gene regions, cytochrome c oxidase I (COI) and 16S rRNA, were conducted using the primers that were listed in Table S1 as follows: initial denaturation at 98 °C for 3 min, followed by 40 cycles at 98 °C for 10 s, 52 °C for 10 s, 72 °C for 10 s, and a final extension at 72 °C for 5 min. The total reaction volume was 50 μL that included 2 μL of template DNA, 1 μL of each primer, and 46 μL of GoldenStar® T6Super PCR Mix (1.1×). PCR products were assessed on GelRed-stained 1.5% agarose gel electrophoresis and sequenced in both directions using the ABI 3730 DNA Analyzer sequencing facility from BGI Genomics, Shenzhen, Guangdong Province, China. The final sequences were deposited in the Science Data Bank of Chinese Academy of Sciences (Xiao & Zhang, 2024) and the GenBank database with the accession numbers PP869369–PP869372 (COI) and PP868346–PP868349 (16S).

Phylogenetic analyses and genetic distance

Molecular phylogenetic analyses were conducted using partial mitochondrial gene seuquences of cytochrome c oxidase I (COI) and 16S rRNA, obtained from each of the four specimens in this study, and some relevant sequences of elasipodid species and two out-group species of the genus Pseudostichopus Théel, 1886 (Holothuroidea, Persiculida, Pseudostichopodidae), which were downloaded from National Center for Biotechnology Information (NCBI). A final total of 36 16S sequences and 128 COI sequences were obtained (Table S2). For the phylogenetic investigation, we prepared the alignment datasets of the COI and the concatenated regions of 16S and COI (16S-COI). The sequences were aligned using MAFFT v.7 (Katoh & Standley, 2013) with default parameters. The evolutionary model GTR+G+I was the best-fitted model for both alignments, which was selected using PartitionFinder2 (Lanfear et al., 2017), with all algorithms and AICc criteria. The analyses were performed on an unpartitioned dataset, where both 16S and COI sequences were treated as a single partition. Maximum Likelihood (ML) analysis was inferred using the Shimodaira-Hasegawa-like approximation, likelihood-ratio test (Gascuel, 2010) and IQ-TREE (Lam-Tung et al., 2015) models with 20,000 ultrafast bootstraps (Minh, Nguyen & Von, 2013). For the ML bootstraps, we considered values <70% as low, 70–94% as moderate, and ≥95% as high, following Hillis & Bull (1993). Bayesian Inference phylogenies (BI) were inferred using MrBayes 3.2.6 (Ronquist et al., 2012) under the partition model (two parallel runs, 5,000,000 generations). The initial 25% of sampled data were discarded as burn-in, and the remaining trees were summarized in a 50% majority rule consensus tree. For the Bayesian posterior probabilities, we considered values <0.95 as low and >0.95 as high, following Alfaro, Zoller & Lutzoni (2003). The results were visualized using FigTree v. 1.4.4 (Rambaut, 2018). We estimated the genetic distances of COI, which were calculated using the Kimura two-parameter model, within each Psychropotes species and among different Psychropotes species using model MEGA X (Kumar et al., 2018).

Analyses of species diversity and distribution

Existing distribution data of psychropotid species were extracted from the Ocean Biodiversity Information System (OBIS) (https://obis.org/) and the Global Biodiversity Information Facility (GBIF) (https://www.gbif.org/zh/), from published literature, and from this study. The distribution of three genera in the family Psychropotidae was illustrated using Generic Mapping Tools, which is a cartographic scripting toolset developed by Wessel & Smith (1995). Using the World Register of Marine Species (WoRMS) and primary literature (e.g., Vaney, 1908; Théel, 1882; Ludwig, 1893; Koehler & Vaney, 1905; Hansen, 1975; Gebruk, 2008; Rogacheva, Cross & Billett, 2009; Li et al., 2018; Xiao, Li & Sha, 2018; Gebruk, Kremenetskaia & Rouse, 2020), we obtained all valid species in the family and discussed the geographical distribution of different groups.

Results

Systematics

Order Elasipodida Théel, 1882

Family Psychropotidae Théel, 1882

Genus Psychropotes Théel, 1882

Diagnosis. “Anus ventral. Unpaired dorsal appendage present. Circum-oral papillae absent. Tentacles discs of a fixed shape, rounded in outline and with marginal knobs” (from Hansen, 1975: 99).

Psychropotes asperatus sp. nov.

(Figs. 2–4)

Figure 2 Psychropotes asperatus sp. nov., holotype (NIWA164003).

(A, B) The holotype specimen in situ. (C, D) The holotype in preserved state. (E) Dorsal warts. (F) Closeup shooting of warts on dorsum. Scale bars: 5 cm (A, B); 2 cm (C, D); 500 μm (E, F).

Figure 3 Psychropotes asperatus sp. nov., SEM, holotype (NIWA164003).

(A–F) Ossicles from dorsal body wall: (A–E) large crosses with arms bent downwards or irregular bending direction; central apophysis long, spinose or smooth; (F) small deposits with three, four, and five arms almost straight, central apophysis short or rudimentary. (G–L) Unpaired dorsal appendage: (G–I, K–L) crosses with single central apophysis, bearing spines or smooth; (J) rods with the branch emanating from the center forming a ‘third arm’. Scale bar: 300 μm.

Figure 4 Psychropotes asperatus sp. nov., SEM, holotype (NIWA164003).

(A–D) Ossicles from tentacles: (A, C) sturdy and rather large rods, terminal part rounded; (B) tripartite deposits; (D) smooth rods. (E, F) tube feet: rods, both ends surrounded by small spines. (G–J) ventral body wall: (G, H) small crosses, arms with spines distributed over at least distal half, central apophysis rudimentary; (I) simple rods; (J) tripartite deposits. Scale bar: 300 μm.

urn:lsid:zoobank.org:act:FF8C7F3A-D0AA-44D3-9C83-7A9D4032D0A2

Material examined. Holotype. NIWA164003, collected from the overriding plate of Kermadec Trench in the South Pacific Ocean, Dive FDZ126 (30°25.51′S, 177°54.60′W), depth 6,018 m, 1 Nov. 2022, preserved in 99% high grade absolute ethanol.

Type locality. The Kermadec Trench, South Pacific Ocean, depth 6018 m.

Diagnosis. Body elongated, dorsum convex, ventrum flattened. Brim narrow. Mouth ventral, anus ventral. Tentacles 12. Dorsal papillae not found. Unpaired appendage about 1/3 of body length, end bifurcated, placed about 1/3 of body length from posterior body end. Skin violet, unpaired appendage and the middle of the dorsum and ventrum often darker. Dorsal skin and unpaired appendage covered with warts. Dorsal deposits two types, large crosses and small deposits with three, four, and five arms. Unpaired dorsal appendage with large crosses. Tentacles and tube feet with rods. Ventral deposits small crosses, rods, and tripartite deposits.

Description. External morphology. Body elongated, dorsum convex, ventrum flattened (Figs. 2A and 2B). 15 cm long and 7.5 cm wide after fixation (Figs. 2C, 2D). Color in life violet, darker in the middle of the dorsum and ventrum. Dorsal skin gelatinous and soft, with a rough surface and conspicuous warts covering body and appendage (Figs. 2C, 2E and 2F), ventral skin smooth (Fig. 2D). Mouth ventral, anus ventral. Tentacle 12, completely retracted into pockets of the skin, with round terminal discs, circum-oral papillae absent. Brim composed by fused tube feet, relatively narrow, retracted when fixed (Figs. 2C and 2D). Midventral tube feet small, evenly distributed and densely packed, arranged along the middle of ventrum. Dorsal papillae not found. The unpaired dorsal appendage end bifurcated (Figs. 2A and 2B) in situ. In preserved state, unpaired dorsal appendage 4.6 cm long, about 1/3 of body length, 2.5 cm at the base, placed about 1/3 of body length from the posterior end (Fig. 2C), covered with warts as on the dorsal surface of the body.

Ossicle morphology. Dorsal deposits with two main types. (1) Large crosses (Figs. 3A–3E) with arms 248–816 μm in length (mean 580 μm, N = 28), bent downwards or irregular bending direction (Fig. 3C), each arm surrounded by numerous small spines; central apophysis long, spinose (Figs. 3A–3D) or smooth (Fig. 3E), 136–375 μm in length (mean 239 μm, N = 28). (2) Small deposits with three, four, and five arms almost straight, central apophysis short or rudimentary (Fig. 3F). Deposits with four arms slender, varying in length from short to long, 57–242 μm in length (mean 140 μm, N = 35), with sparse or conspicuous spines (sometimes bifurcated); tripartite deposits with conspicuous spines, proximal spines often bifurcated, three arms almost equal in length, each arm 88–162 μm long (mean 133 μm, N = 24), central apophysis short or reduced; quinquepartite deposits, five arms almost equal in length, each arm 76–228 μm long (mean 140 μm, N = 34), central apophysis branched or rudimentary. Crosses and rods found in unpaired dorsal appendage (Figs. 3G–3L). Crosses with single central apophysis, 147–213 μm in length (mean 163 μm, N = 20), bearing spines (Figs. 3H, 3I, 3K and 3L) or smooth (Fig. 3G), arms bent downwards from central apophysis, 238–500 μm in length (mean 345 μm, N = 20), sometimes with irregular curvature; rods up to 427 μm long (Fig. 3J), with the branch emanating from the center forming a ‘third arm’, only the ends possessing spines. Tentacles with three types of rods (Figs. 4A–4D). (1) Sturdy and rather large rods (Figs. 4A and 4C), 706–1,300 μm long (mean 1,009 μm, N = 25), terminal part rounded, bearing spines. (2) Tripartite deposits (Fig. 4B), each arm 383–505 μm long (mean 442 μm, N = 15). (3) Smooth rods (Fig. 4D), 202–462 μm in length (mean 320 μm, N = 16). Deposits of tube feet rods (Figs. 4E and 4F), 337–640 μm in length (mean 488 μm, N = 25), ends of rods surrounded by few small spines. Simple rods, tripartite deposits and crosses in ventral body wall (Figs. 4G–4J), rods (Fig. 4I) same as those in tube feet; tripartite deposits (Fig. 4J) similar to those in tentacles, but smaller, each arm about 62–160 μm in length (mean 129 μm, N = 10); arms of smaller crosses 31–185 μm in length (mean 93 μm, N = 25), with spines distributed over at least distal half, central apophysis rudimentary (Figs. 4G and 4H).

Etymology. The species name is derived from the Latin word asperatus, which means rough, and this refers to the rough dorsal skin caused by warts.

Distribution. Only known in the type locality.

Remarks. Psychropotes asperatus sp. nov. clearly belongs to the genus Psychropotes, and it differs from other congeners by the tripartite (mean arm length 133 μm, N = 24 from one individual) and quinquepartite (mean arm length 140 μm, N = 34 from one individual) deposits on the dorsum.

Three species of Psychropotes are characterized by warts on their dorsal skin (i.e., P. verrucicaudatus, P. verrucosa, and P. mirabilis) (Ludwig, 1893; Hansen, 1975; Xiao et al., 2019). Among these, the dorsal appendages of P. verrucosa and P. mirabilis are smooth, whereas those of P. asperatus sp. nov. and P. verrucicaudatus are covered with warts, this made the new species most similar to P. verrucicaudatus, which was described by Xiao et al. (2019) based on one specimen collected from the South China Sea. Morphologically, the new species differed from P. verrucicaudatus by the length and position of the dorsal appendage, and there were some differences in ossicle characters. In P. verrucicaudatus, the appendage was very short, measured about 1/12 of the body length, placed about 2/5 of the body length from the posterior end of the body, but in the new species, the appendage was about 1/3 of the body length, and it was placed about 1/3 of the body length from the posterior end of the body. In ossicle morphology, Psychropotes asperatus sp. nov. had large crosses with arms of 248–816 μm, while the giant crosses of P. verrucicaudatus had arms 600–750 μm in length, none of the crosses with arm lengths shorter than 600 μm was observed in P. verrucicaudatus. In addition, in P. verrucicaudatus, the central apophysis of giant crosses bear spines, but in the new species, the crosses had a central apophysis that was spinous or smooth. The new species was differentiated from P. verrucicaudatus by possessing tripartite and quinquepartite deposits on the dorsum in addition to crosses. For the rods in the tentacles, those of P. verrucicaudatus were 210–600 μm long with multiple and irregularly arranged spines, whereas the rods of the new species were more robust, 706–1,300 μm long, and with spines only at the ends.

Psychropotes diutiuscauda sp. nov.

(Figs. 5–8)

Figure 5 Psychropotes diutiuscauda sp. nov., holotype (IDSSE-EEB-HS175).

(A, B) In situ images. (C) Specimen before fixation in absolute ethanol. (D, E) Preserved in absolute ethanol after a few days. Scale bars: 10 cm (A, B); 3 cm (C–E).

Figure 6 Psychropotes diutiuscauda sp. nov., SEM, holotype (IDSSE-EEB-HS175).

Ossicles from dorsal body wall: (A–C) irregular crosses in inner layer, with extremely slender arms, small unbranched spines alternately spaced above and below on arms; (D) quinquepartite deposits, bearing irregular spines, proximal spines larger than others, bifurcate or tripartite, central apophysis replaced by multiple spines; (E–I) crosses, arms usually with a horizontal curvature and bearing irregular or secondary spines. Scale bar: 100 μm.

Figure 7 Psychropotes diutiuscauda sp. nov., SEM, holotype (IDSSE-EEB-HS175).

Ossicles from ventral body wall: (A–C) quinquepartite deposits with single central apophysis or reduced; (D–I) crosses with single central apophysis; (J, K) crosses with reduced apophysis; (L, M) crosses with two central apophysis; (N) crosses with central tripartite apophysis; (O, P) relatively slender crosses, single central apophysis spinous. Scale bar: 50 μm.

Figure 8 Psychropotes diutiuscauda sp. nov., SEM, holotype (IDSSE-EEB-HS175).

(A–Q) Ossicles from tube feet: (A–F) arms of crosses densely distributed with conspicuous spines, proximal large spines often bifurcate or tripartite, sometimes bearing irregular secondary spines, single central apophysis relatively short, equal in length to proximal large spines; (G–I) crosses with rather long apophysis, almost equal to the length of the arms, forming a “fifth arm”; (J, K) crosses with two short or two long central apophysis; (M–Q) crosses with slender arms central apophysis often reduced. (R–V) Ossicles from tentacles: (R–U) rather large rods; (V) smaller crosses and rods with an extra branch from the center. Scale bars: 100 μm.

urn:lsid:zoobank.org:act:E7FE330B-DA67-4D70-B1F6-6CA468E47874

Materials examined. Holotype. IDSSE-EEB-HS175, collected from the WZFZ in the East Indian Ocean, Dive FDZ188 (22°23′S, 102°27′E), depth 6,397 m, 25 Feb. 2023, preserved in 99% high grade absolute ethanol.

Type locality. WZFZ, East Indian Ocean, depth 6,397 m.

Diagnosis. Body elongated, color yellowish green. Tentacles 15. Brim broad on anterior end of the body. Ventrolateral tube feet up to nine pairs. Midventral tube feet small, arranged in two rows. Unpaired appendage rather large, about 1.5 times as long as body length, placed close to posterior end of the body. Dorsal deposits cross-shaped, quinquepartite. Tentacles with rather large rods and smaller crosses. Tube feet crosses with single and two central apophysis, short or longer than the arms. Ventral deposits irregular rods, crosses, and quinquepartite deposits, single or two central apophysis on under side of the crosses.

Description. External morphology. Body elongated, convex dorsally and flattened ventrally (Figs. 5A–5C), about 17 cm long and 7 cm wide in situ, 9 cm in maximum width in the anterior apart in situ. Unpaired dorsal appendage rather large, 24 cm long in situ, 1.5 times the length of the body (Figs. 5A–5C), place close to posterior end of the body. In preserved state, body 9 cm in length and 4 cm in width (Figs. 5D and 5E), unpaired dorsal appendage rather large and long, 11.5 cm in length, and as rough as the dorsal skin. Color yellowish green (Figs. 5A–5E), skin rough. Tentacles 15, shield-shaped. Brim well developed at anterior end of the body (Fig. 5A). Ventrolateral tube feet approximately nine pairs, arranged along the entire edge of the brim. Dorsal papillae not found. Midventral tube feet small, evenly distributed in two rows along ventral sole; the number unclear due to damaged ventral body wall in holotype (after preservation).

Ossicle morphology. Dorsal deposits crosses and quinquepartite. (1) Irregular crosses in inner layer of body wall, with extremely slender arms (Figs. 6A–6C), 70–160 μm in length (mean 104 μm, N = 10), small unbranched spines alternately spaced above and below on arms. (2) Quinquepartite deposits (Fig. 6D), each arm 69–258 μm in length (mean 121 μm, N = 14), bearing irregular spines, proximal spines larger than others, bifurcate or tripartite, central apophysis replaced by multiple spines. (3) Arms of crosses, usually with a horizontal curvature (Figs. 6E–6I), 54–215 μm in length (mean 117 μm, N = 30). The spines numerous and large, tapering toward the end of the arms, with irregular or secondary spines, central apophysis replaced by branched spines. Deposits on ventrum crosses and quinquepartite (Figs. 7A–7P). Quinquepartite deposits with single central apophysis or reduced, each arm 37–90 μm in length (mean 54 μm, N = 22) (Figs. 7A–7C), bearing spines. Crosses of two types: (1) Arms varing from almost straight to sharply bent (Figs. 7D–7N), 30–90 μm in length (mean 59 μm, N = 35), with prominent irregular spines. Four types of central apophysis: single (Figs. 7D–7I), 13–29 μm in length (mean 20 μm, N = 13); reduced (Figs. 7J, 7K); two central apophysis (Figs. 7L and 7M) on under side of the crosses, 7–30 μm in length, (mean 27 μm, N = 7); or apophysis tripartite (Fig. 7N), 10–20 μm in length (mean 12 μm, N = 5), bearing branched spines. (2) Relatively slender crosses (Figs. 7O and 7P), arms 40–140 μm in length (mean 86 μm, N = 24), at least half of the arms with small spines, single central apophysis spinous. Deposits on tube feet with three types of crosses, 80–244 μm in length (mean 150 μm, N = 38) (Figs. 8A–8Q). (1) Arms of crosses densely distributed with conspicuous spines, proximal spines larger than others, often bifurcate or tripartite, sometimes bearing irregular secondary spines. Single central apophysis relatively short, equal in length to proximal large spines (Figs. 8A–8F), or rather long, almost equal to the length of the arms (Figs. 8G–8I), 79–294 μm in length (mean 164 μm, N = 12), forming a “fifth arm”, apophysis also spinous. (2) Crosses with two short or two long central apophysis (Figs. 8J and 8K), bearing spines. (3) Crosses with slender arms (Figs. 8M–8Q), 126–200 μm in length (mean 174 μm, N = 18), central apophysis often reduced. Rather large rods and smaller crosses in tentacles (Figs. 8R–8V), rods 429–1,000 μm in length (mean 739 μm, N = 20), some with an extra branch from the center (Figs. 8R and 8V), bearing irregular small spines; crosses with spinose arms 40–110 μm in length (mean 66 μm, N = 25) (Fig. 8V), conspicuous spines distributed over at least distal half, central apophysis rudimentary.

Etymology. From Latin ‘diutius’ for longer, and ‘cauda’, meaning tail. The name refers to its dorsal appendage that was longer than its body length.

Distribution. Only known in the type locality.

Remarks. The new species clearly belongs to Psychropotes, the unpaired dorsal appendages of ‘Psychropotes longicauda’ species were usually 1/5–1/1 length of the body, whereas the appendages of the new species were about 1.5 times the length of the body. Psychropotes diutiuscauda sp. nov. differed from other species of Psychropotes by the presence of quinquepartite deposits on the dorsum (mean arm length 121 μm, N = 14 from 1 individual) and ventrum (mean arm length 54 μm, N = 22 from 1 individual), and ventral crosses were strongly spinous, with reduced, tripartite, single or two central apophyses developed on the underside of the crosses.

The unique yellowish-green color made P. diutiuscauda sp. nov. most similar to P. xenochromata, P. dyscrita, and P. moskalevi (Clark, 1920; Rogacheva, Cross & Billett, 2009; Gebruk, Kremenetskaia & Rouse, 2020). Compared with the 18 tentacles of the other three species, the number of tentacles of the new species was fewer, only 15. The length of unpaired dorsal appendage was 1.5 times of the body length in the new species, but the unpaired dorsal appendages were 1/5–1/2 of the body length in P. xenochromata, 1/5–1/1 of the body length in P. moskalevi, and 4/5 of the body length in P. dyscrita. Among the yellowish-green species, P. diutiuscauda sp. nov. matched well with P. moskalevi with its dorsal deposits of horizontally curved arms, short spines on the arms, with proximal spines often bipartite or bearing secondary spines, which were about the same length as the central apophysis. However, the new species differed from P. moskalevi by possessing quinquepartite deposits on the dorsum and ventrum, by the absence of tripartite spines and rods on the ventrum, and by ventral crosses that were strongly spinous, with single, reduced, and two central apophyses, which were developed on the underside of the crosses.

Psychropotes nigrimargaria sp. nov.

(Figs. 9–11)

Figure 9 Psychropotes nigrimargaria sp. nov., holotype (IDSSE-EEB-HS176).

(A, B) In situ images. (C) Specimen before fixation. ( D, E) Specimen after preservation. Scale bars: 10 cm (A, B), 3 cm (C–E).

Figure 10 Psychropotes nigrimargaria sp. nov., SEM, holotype (IDSSE-EEB-HS176).

(A–G) Ossicles from dorsal body wall: arms of crosses bearing 1–3 proximal spines, the largest spines often bipartite or irregular branched, equal in length to central apophysis; (D, G) central apophysis tripartite or quadripartite. (H–J) Ossicles from ventral body wall: relatively smaller crosses with fewer spines, central apophysis reduced or short. Scale bars: 100 μm.

Figure 11 Psychropotes nigrimargaria sp. nov., SEM, holotype (IDSSE-EEB-HS176).

(A–K) Ossicles from tube feet: (A–C, E–G) smaller crosses similar to those in dorsal body wall; (I–K) smaller crosses similar to those in ventral body wall; (D) crosses and tripartite deposits with spines at the ends; (H) robust rods and tripartite deposits. (L–N) Ossicles from tentacles: (L) crosses bearing small spines or smooth, irregularly spinous central apophysis short or reduced; (M) simple rods smooth or spinous, some possessing an extra branch in middle; (N) robust rods, spinous ends branched, and smooth rods. Scale bar: 100 μm.

urn:lsid:zoobank.org:act:2C252D0F-16BA-4E6E-9F17-D8AF01D43CB8

Materials examined. Holotype. IDSSE-EEB-HS176, collected from the WZFZ in the East Indian Ocean, Dive FDZ188 (22°22′S, 102°27′E), depth 6,605 m, 25 Feb. 2023, preserved in 99% high grade absolute ethanol.

Type locality. WZFZ, East Indian Ocean, depth 6,605 m.

Diagnosis. Body elongated, up to 30 cm long. Color dark violet, tentacles, and tube feet darker. Tentacles 18, with round terminal discs. Brim broad, especially at anterior and posterior ends. Free ventrolateral tube feet 12 pairs. Unpaired dorsal appendage about 1/3 of body length placed some distance from the posterior end. Dorsal deposit crosses, central apophysis tripartite or quadripartite. Crosses and rods in tentacles. Tube feet deposits crosses, tripartite, and rods. Ventral crosses smaller than the dorsal ones.

Description. External morphology. Body elongated, about 30 cm long and 10 cm wide in situ (Figs. 9A and 9B). In preserved state, body approximately 16.5 cm in length and 8 cm in width (Figs. 9D and 9E). Color dark violet, tentacles, and tube feet darker, almost black. Skin thick. Tentacles 18, with round terminal discs (Figs. 9C and 9E). Brim well developed especially at anterior and posterior ends of the body, with maximum width around anterior end of 11 cm (Figs. 9A and 9B). Free ventrolateral tube feet 12 pairs, retracted into the body after fixation. The mid-ventral tube feet in poor condition, and not clearly observed. Dorsal papillae not found. Unpaired dorsal appendage conspicuous and conical, tapering toward the end, in situ 10 cm long, about 1/3 of the body length (Figs. 9A and 9B); in preserved state, 6 cm long, developed at 3 cm from posterior body end (Figs. 9D and 9E).

Ossicle morphology. Dorsal deposits crosses with arms 80–310 μm in length (mean 202 μm, N = 33) (Figs. 10A–10G), sharply bent downwards, arm spines conspicuous; bearing 1–3 proximal spines, the largest spines often bipartite or irregular branched, their length equal to that of central apophysis, additional smaller spines on arm ends. Central apophysis tripartite (N = 12) or quadripartite (N = 17) (Figs. 10D–10G). Ventral crosses smaller than the dorsal ones, with arms 40–154 μm in length (mean 117 μm, N = 36), bearing fewer spines, and central apophysis reduced or short (Figs. 10H–10J). Tube feet crosses, tripartite deposits, and rods (Figs. 11A–11K), crosses with arms 83–267 μm long (mean 152 μm, N = 33), similar to dorsal crosses (Figs. 11A–11C and 11E–11G) and ventral crosses (Figs. 11I–11K) but smaller; robust rods 313–416 μm long (mean 348 μm, N = 21) (Fig. 11H); tripartite deposits with spines at the ends (Fig. 11D and 11H). Crosses and rods in tentacles (Figs. 11L–11N), crosses with arms 60–160 μm in length (mean 99 μm, N = 20), bearing small spines or smooth, irregularly spinous central apophysis short or reduced (Fig. 11L); robust and simple rods 247–606 μm in length (mean 419 μm, N = 33), smooth or spinous ends branched (Figs. 11M and 11N), some possessing an extra branch in the middle (Fig. 11M).

Etymology. From Latin, nigri, black, and margaria, pearl, meaning “black pearl”, which alludes to the shape and color of the tentacles of this species.

Distribution. Only known in the type locality.

Remarks. The new species belonged to the genus Psychropotes by the presence of unpaired dorsal appendage and ventral anus. Arms of dorsal crosses bear large proximal spines, bipartite or branched irregularly, and the length of proximal spines were equal to that of the central apophysis, which made the new species most similar to P. raripes and P. monstrosa (Théel, 1882; Ludwig, 1893; Gebruk, Kremenetskaia & Rouse, 2020).

However, Psychropotes nigrimargaria sp. nov. differed from P. raripes by the absence of dorsal papillae, P. raripes has 5–7 pairs. The number of free ventrolateral tube feet in P. raripes was 7–10 pairs, but the new species had 12 pairs. Unpaired dorsal appendages were up to 3/4 of the body length in P. raripes and placed close to the posterior end of the body, but in the new species they were about 1/3 of the body length and developed at ~1/5 of the body length from posterior end. For ossicle morphology, P. nigrimargaria sp. nov. had dorsal crosses with central apophyses that were tripartite (N = 12) and quadripartite (N = 17) and ventral small crosses with fewer spines on the arms, while P. raripes had tripartite deposits, and dorsal crosses with central apophyses that were tripartite; ventral crosses tripartite, and star-shaped deposits with conspicuous spines on the arms.

P. nigrimargaria sp. nov. differed from P. monstrosa by the absence of dorsal papillae, P. monstrosa had 5–7 pairs. The number of free ventrolateral tube feet in P. monstrosa was 18–20 pairs, but the new species had 12 pairs. The unpaired dorsal appendage was almost 1/2 of the body length in P. monstrosa, and it was about 1/3 of the body length in the new species. For ossicle morphology, the new species differed from P. monstrosa by the absence of dorsal deposits with strongly arcuate arms, and the presence of tripartite and quadripartite central apophyses that were unbranched in P. monstrosa.

Psychropotes depressa (Théel, 1882)

(Figs. 12 and 13)

Figure 12 Psychropotes depressa (Théel, 1882) (NIWA164160).

(A) In situ images. (B) Specimen after fixation. (C–J) Dorsal ossicles with four proximal spines under SEM. Scale bars: (A) 10 cm; (B) 1 cm; (C–J) 200 μm.

Figure 13 Psychropotes depressa (Théel, 1882), drawing and SEM, (NIWA164160).

(A) Ossicles from dorsal papillae: crosses with arms sharply curved downward and then upward at the end, bearing smaller proximal spines. (B) Ossicles from dorsal body wall: crosses with slender arms, bent straight down. (C) Ossicles from unpaired dorsal appendage similar to those on dorsal body wall. (D) Ossicles from tentacles: curved rods. Scale bars: 100 μm.

Euphronides depressa Théel, 1882, pp. 93–96, pl. XXVI, XXXV, XL, XLVI; Ohshima, 1915, pp. 244–245, fig.1.

Psychropotes depressa Hansen, 1975, pp. 106–111, fig. 43, 44, pl. VII, XII, XIV; Gebruk, 2008, pp. 50–51; Rogacheva, Gebruk & Alt, 2013, pp. 599, figs. 17f, 17g; Xiao, Li & Sha, 2018, pp. 5–10, fig. 6–10.

Euphronides depressa var. minor Théel, 1886, p. 2.

Euphronides cornuta Verrill, 1884, p. 217; Veriill, 1885, pp. 518, 538, fig. 32, 33; Deichmann, 1930, pp. 127–128; Heding, 1940, p. 368.

Euphronides tanneri Ludwig, 1894, pp. 39–44, pl. III, IV, V.

Euphronides auriculata Perrier, 1896, pp. 901–902; Perrier, 1902, pp. 434–438, pl. XIII, XX; Grieg, 1921, pp. 8, 9.

Euphronides violacea Perrier, 1896, p. 902; Perrier, 1902: 438–441, plate XX; Deichmann, 1930, pp. 128–129; Deichmann, 1940, pp. 201–202; Heding, 1942, pp. 15–16; Madsen, 1947, p. 16; Deichmann, 1954, p. 384.

Euphronides talismani Perrier, 1896, p. 902; Perrier, 1902, pp. 441–444, plate XX; Hérouard, 1902, pp. 30–31, plate II; Deichmann, 1930, p. 129; Heding, 1942, p. 15, fig. 15.

Benthodytes assimilis Théel, 1886, pp. 2–3.

Material examined. NIWA164160, collected from the Kermadec Arc in the South Pacific Ocean, Dive FDZ150 (27°34.54’S, 177°8.51’W), depth 1620 m, 8 Dec. 2022, preserved in 99% high grade absolute ethanol.

Description. External morphology. The specimen 27 cm long and 9.5 cm wide in situ (Fig. 12A), color violet, slightly darker on ventrum (Fig. 12B). After several days of fixation in 99% high grade absolute ethanol, body flattened, 9 cm long and 4 cm maximum width in anterior part (Fig. 12B). Tentacle 12, round terminal discs with short, digitiform projections on the margin. Brim broad. Midventral tube feet small, arranged in two rows. Dorsal papillae five pairs (Figs. 12A and 12B), minute, gradually increasing the interval and size; the fifth pair of papillae largest, 9 mm in length, and placed about 1/2 of the body from posterior end. In preserved state (Fig. 12B), unpaired appendage 2.2 cm long, and 2 cm wide at the base, placed in 2.7 cm (about 1/3 of body length) from the posterior end.

Ossicle morphology. Dorsal crosses with slender arms (Figs. 12C–12J and 13B), long and smooth central apophysis, arms 144–395 μm in length (mean 269 μm, N = 28), central apophysis 75–123 μm in length (mean 89 μm, N = 17), bent downwards proximally, and slightly curved upwards or horizontal in the distal part. Arms often with four proximal spines, about 3/5 of the length of the central apophysis (Figs. 12C, 12D, 12G and 12I), or shorter than central apophysis but larger than other spines (Figs. 12E and 12J). After repeated examinations, ventral deposits not found. Dorsal papillae crosses (Fig. 13A), arms 142–276 μm in length (mean 205 μm, N = 22), sharply curved downward and then upward at the end, bearing smaller proximal spines. Deposits of the unpaired appendage similar to those on dorsum (Fig. 13C). Tentacles with simple rods (Fig. 13D), 255–450 μm in length (mean 379 μm, N = 20).

Distribution. This species is common throughout the North Atlantic and South Atlantic, and Pacific Ocean (Japan, Caroline seamount, Gulf of Panama, Chile, and the Kermadec Arc). Depth 957–4,200 m (Hansen, 1975; Xiao, Li & Sha, 2018).

Remarks. This species has a wide geographical distribution and exhibited a high degree of variability in its morphological characteristics (Hansen, 1975; Xiao, Li & Sha, 2018). There have been several synonyms (Théel, 1882; Verrill, 1884; Théel, 1886; Perrier, 1896; Hérouard, 1902; Perrier, 1902; Deichmann, 1930), and the synonymous species were distinguished from each other by the shape of the body, the size of the posterior-most pair of dorsal papillae, the size of the unpaired appendage, body color, and the size and shape of deposits. By re-examining the specimen and comparing descriptions in the literature, Hansen (1975) determined that none of these differences were valid identifying characters and that there were age variations in some features, such as the number of tentacles, skin texture, the size of the posterior-most pair of dorsal papillae; geographic variation may cause changes in certain features, such as the shape and size of crosses and whether or not the long central apophysis bear spines.

The morphological features of the specimen in this study, which included the position and length of the unpaired dorsal appendage and the types of ossicles, were consistent with those of P. depressa. However, the following differences in morphological features occurred: (1) five pairs of dorsal papillae, (2) deposits not found on ventrum, (3) the posterior-most pair of papillae were larger than the others, and (4) only 12 tentacles instead of the usual 18 tentacles. Our specimen conformed to the known range of variation in P. depressa that was described by Hansen (1975), Ohshima (1915), and Xiao, Li & Sha (2018). Therefore, we identified it as P. depressa. This was the first record of P. depressa from the Kermadec Trench. In addition, we supplemented a description of the dorsal papillae deposits (Fig. 6A), which were more similar to the original description of the dorsal surfaced crosses, but this trait was not mentioned in the original literature. The arms were bent sharply downward, then upward at the ends, and arms bear smaller spines near the central apophysis.

Key to the species of Psychropotes Théel, 1882 (adapted from Xiao et al., 2019) 1. Body transparent, colorlessP. hyalinus

Body non-transparent, colored2

2. Body yellowish or yellow-greenish3

Body violet or purple6

3. Tentacles 17–20, the appendage is 1/5–1/1 of body length4

Tentacles 15, the appendage is 1.5 times of body lengthP. diutiuscauda sp. nov.

4. With free ventrolateral tube feet5

Without free ventrolateral tube feetP. xenochromata

5. Ventrolateral tube feet 13–15 pairs, dorsal papillae absentP. dyscrita

Ventrolateral tube feet up to 25 (often 17–19) pairs, dorsal papillae up to 8 pairsP. moskalevi

6. Dorsal appendage placed close to posterior end of body7

Dorsal appendage placed at least 1/5 body length from posterior end of body13

7. Tentacles 10–128

Tentacles 189

8. Ventral crosses with a low and spinous central apophysis, or with no apophysis, arms up to 0.2 mm long, with small spinesP. loveni

Ventral crosses with long arms (up to 0.3 mm) and long central apophysis, without spinesP. dubiosa

9. Dorsal papillae present10

Dorsal papillae absentP. nigrimargaria sp. nov.

10. Relatively a low number of ventrolateral tube feet, 7–11 pairs11

Relatively a high number of ventrolateral tube feet, up to 26 pairs12

11. Anterior brim consists of about 18–20 tube feet, and posterior brim of 13–14 tube feetP. raripes

Anterior brim consists of about 13–17 tube feet, and posterior brim of 3–6 tube feetP. buglossa

12. Dorsal ossicles with arms only of short type, 0.06–0.1 mm in length; dorsal papillae small, 4 pairsP. fuscopurpurea

Dorsal ossicles with arms of both short and long types, 0.06 mm or 0.24–0.4 mm in length; dorsal papillae minute, 5 pairsP. longicauda

13. Dorsal skin covered with warts14

Dorsal skin smooth15

14. Dorsal appendage covered with warts16

Dorsal appendage smooth17

15. Dorsal appendage at the most 1/6 length of the body18

Dorsal appendage at least 1/3 length of the body19

16. The appendage is very short, measured about 1/12 of body length, placed about 2/5 of body length from posterior end of bodyP. verrucicaudatus

The appendage is 1/3 of body length and placed about 1/3 of body length from posterior end of bodyP. asperatus sp. nov.

17. Dorsal appendage shortP. verrucosa

Dorsal appendage very longP. mirabilis

18. Tentacles 18P. depressa

Tentacles 16P. scotiae

19. Brim broad. Dorsal appendage 1/3 to 1/2 length of the body20

Brim narrow. Dorsal appendage 1/3 to 1/1 length of the body21

20. Dorsal papillae up to 3 pairs, dorsal crosses with small spines placed in rings on the arms, central apophysis rudimentary or absent…P. belyaevi

Dorsal papillae 5–7 pairs, dorsal crosses with strongly bent arms, bearing two proximal spines, central apophysis largeP. monstrosa

21. Tentacles 15–16, without free ventrolateral tube feet22

Tentacles 18, free ventrolateral feet 20–21 pairsP. pawsoni

22. Two types of ossicles: one with spines throughout arm length, the other with a smooth, proximal arm part and a high, central apophysis ending in three or four downwardly bent hooksP. semperiana

Only one type of ossicles: small, slender crosses with a low, central apophysisP. minuta

Genetic distances and phylogenetic analyses

A total of four COI sequences and four 16S sequences from three new species and one new record were deposited into GenBank (Table S2). The COI sequences from ten species of Psychropotes were used to perform pairwise, uncorrected p-distance analyses based on the Kimura two-parameter (K2P) model (Table 1). For COI alignment, the interspecific distances ranged between 1.38–13.54%, and the intraspecific distances ranged from 0.00 to 1.23% (Table 1). The alignment of the total of 19 sequences contained a character matrix consisting of 482 nucleotide sites, with 70 variable sites and 58 parsimony-informative sites.

Table 1 Interspecific and intraspecific uncorrected p-distances at COI of ten species of Psychropotes.

	1	2	3	4	5	6	7	8	9	10	
1. P. asperatus sp. nov.	–										
2. P. depressa	9.87–9.99%	1.07%									
3. P. diutiuscauda sp. nov.	11.33%	11.67–12.05%	–								
4. P. dyscrita	13.05%	12.48–12.81%	2.55%	–							
5. P. longicauda	11.12%	11.43–11.69%	1.88%	2.73%	–						
6. P. moskalevi	10.22–11.23%	11.68–13.06%	1.56–2.27%	1.38–2.64%	2.32–3.34%	0.00–0.82%					
7. P. nigrimargaria sp. nov.	10.41%	12.01–12.02%	2.84%	4.47%	4.14%	3.48–4.40%	–				
8. P. pawsoni	10.27%	12.21–12.96%	4.30%	6.21%	5.43%	4.99–5.24%	4.52%	–			
9. P. raripes	12.21%	12.80–13.01%	2.47%	2.77%	2.28%	2.47–3.00%	4.85%	4.31%	–		
10. P. verrucicaudatus	3.14–3.26%	10.63–11.67%	11.70–12.65%	12.99–13.54%	10.86–12.62%	10.46–12.66%	11.49–12.38%	11.02–11.35%	12.83–12.85%	0.22–1.23%	
Note:

Intraspecific distances are in bold. ‘–’ means no data.

We conducted the molecular phylogenetic analysis of the order Elasipodida based on all available sequences of four families, and their sampling information and accession numbers are provided in Table S2. Finally, a total of 132 COI sequences trimmed to 481 bp and 40 16S sequences trimmed to 507 bp were obtained after removing non-homologous sites from the sequence alignments, and these were used to reconstruct the BI and ML trees. The four families were grouped together and formed a clade in our phylogenetic analyses, and the topological structures of the ML and BI trees were consistent with the traditional classification system. Tree topologies showed that Elpidiidae, Pelagothuriidae, and Psychropotidae were monophyletic, and Laetmogonidae was polyphyletic. For Laetmogonidae, the molecular evidence was inconsistent with the morphological taxonomy. Compared with phylogenetic trees based solely on the 481 bp mitochondrial COI gene, those trees that were based on concatenated 16S-COI sequences of length 988 bp provided a higher resolution and nodal support (Figs. 14, 15 and Figs. S1–S3).

Figure 14 Maximum likelihood (ML) and Bayesian inference (BI) phylogenetic trees based on concatenated 16S-COI sequences that show phylogenetic relationships among elasipodid species.

Numbers near branches indicate the bootstrap values (BS) from ML (left) and posterior probabilities (PP) of BI (right), and the ‘-’ indicates that the branch is not supported by the BI tree. The thin-annotated branches represent the differences in topologies of the ML and BI trees. The branches of different families are represented by different colors. The sequences of three new species and one new record of Psychropotes provided in this study are in red. BS values < 50 and PP values < 0.5 are not displayed.

Figure 15 Maximum likelihood (ML) and Bayesian inference (BI) phylogenetic trees based on COI sequences that show phylogenetic relationships among psychropotid species.

(A) ML tree, with bootstrap values (BS) labelled. (B) BI tree, with posterior probabilities (PP) labelled. The thin-annotated branches of the ML and BI trees represent the differences in their topologies. The sequences of three new species and one new record of Psychropotes provided in this study are in red. BS values < 50 and PP values < 0.5 are not displayed.

In the 16S-COI trees (Fig. 14 and Fig. S1), the phylogenetic relationships among species of the order Elasipodida were shown as follows. For Laetmogonidae, species of Laetmogone formed a monophyletic clade with full values (BS = 100, PP = 1). Two species in Benthogone each formed an independent clade with low supported values, the genera Psychronaetes and Pannychia were monophyly with full values (BS = 100, PP = 1). The only species in the family Pelagothuriidae, Enypniastes eximia Théel, 1882, clustered with Psychronaetes isolates (BS = 80, PP = 0.8), then followed by Benthogone abstrusa (BS = 64, PP = 0.76). Benthogone rosea and Pannychia species formed a sister clade with low support (BS = 61). For Elpidiidae, the phylogenetic reconstructions of the ML tree showed quite similar topologies compared to the BI tree (Fig. 14 and Fig. S1). Peniagone species formed a sister group with the clade that included genera Amperima + Protelpidia + Scotoplanes + Elpidia (BS = 100, PP = 1). The monophyly of the genera Peniagone and Elpidia were well supported (BS = 100, PP = 1). For the other three genera, the monophyly of Amperima, Protelpidia, and Scotoplanes could not be verified because we had molecular data for only one species in each genus. For Psychropotidae, Benthodytes was paraphyletic and divided into two clades (Clade I: BS = 94, PP = 0.95; Clade II: BS = 100, PP = 1), which was inconsistent with the traditional classification system. The three new species and one new record from the Kermadec Trench fell into the genus Psychropotes, which was a monophyletic group in phylogenetic trees (BS = 98, PP = 0.99). However, the ML and BI trees exhibited somewhat different topological structures for the genus Psychropotes. In the ML tree, Psychropotes divided into two portions (Fig. 14): Portion 1: P. diutiuscauda sp. nov. formed an independent clade, which then clustered with P. dyscrita + P. moskalevi + P. longicauda + P. raripes, followed by P. nigrimargaria sp. nov, they formed a sister clade to the group that included P. longicauda + P. pawsoni; Portion 2: Psychropotes depressa was supported to be a sister to the group comprising P. cf. semperiana + Psycheotrephes exigua (the only species of Psycheotrephes) + P. asperatus sp. nov. + P. verrucicaudatus (BS = 64). While in the BI tree (Fig. S1), P. depressa independently formed a sister clade with all other congeners in Psychropotes (PP = 1); P. diutiuscauda sp. nov. and P. nigrimargaria sp. nov. formed an independent clade, respectively.

In addition to the phylogenetic trees reconstructed based on concatenated 16S-COI sequences, we expanded our analyses by incorporating additional COI sequences reported by Gubili et al. (2017) and conducting phylogenetic analyses focusing solely on COI loci, in order to further verify the accuracy of morphological identification of these species we described. Here, we focus mainly on the phylogenetic relationships among Psychropotes species (Fig. 15 and Figs. S2, S3). The topologies derived from the two different methods, maximum likelihood (ML) and Bayesian inference (BI), were largely consistent (Figs. S2, S3), with only minor variations. Both approaches robustly supported the three new species and the new record distinct from any related genera and species (Figs. 15 and Figs. S2, S3). In the COI trees, Psychropotes depressa independently formed a sister clade with all other congeners in Psychropotes (BS = 99, PP = 1); P. nigrimargaria sp. nov. grouped with Psychropotes sp. 70 and Psychropotes sp. 71 (BS = 100, PP = 1), which then clustered with P. diutiuscauda sp. nov. (BS = 66, PP = 0.82); P. asperatus sp. nov. clustered with P. verrucicaudatus and the only species of Psycheotrephes: P. exigua Théel, 1882 (BS = 95, PP = 1).

Discussion

Generic assignment, species delineation, and taxonomic characters

Both the morphology and molecular phylogenetic analyses supported the assignment of the three new species to the genus Psychropotes. The external morphological features (i.e., absence of ventral anus and circum-oral papillae) in Psychropotes species were most similar to species of Psycheotrephes. However, Psychropotes had unpaired dorsal appendages, which were absent in Psycheotrephes. The three new species in this study conformed to this unique feature, so they all belonged to Psychropotes. P. asperatus sp. nov., P. diutiuscauda sp. nov., and P. nigrimargaria sp. nov. were separated from other congeners by dorsal ossicle types, the length and position of unpaired dorsal appendages, and the number of ventrolateral tube feet.

The discovery of two new species, Psychropotes diutiuscauda sp. nov., and P. nigrimargaria sp. nov., increased the number of the psychropotid holothuroids with long dorsal appendages placed close to or at a short distance from the posterior body (‘P. longicauda’ morphotype) end to 11 species: P. buglossa, P. diutiuscauda sp. nov., P. dubiosa, P. dyscrita, P. fuscopurpurea, P. longicauda, P. monstrosa, P. moskalevi sp. nov., P. nigrimargaria sp. nov., P. pawsoni sp. nov. and P. raripes (Gebruk, Kremenetskaia & Rouse, 2020). Psychropotes diutiuscauda sp. nov. was the fourth reported yellowish-green psychropotid holothuroid besides P. dyscrita, P. moskalevi, and P. xenochromata. Psychropotes xenochromata was discovered as the first yellow-green species, which was unusual for this genus. However, it was later found that the species in the genus Psychropotes, especially the long-tailed species, had diverse colors, and the colors of living specimens ranged from purple/violet to yellow/green/brown (Gebruk, Kremenetskaia & Rouse, 2020).

Thus, even in the living state, only preliminary identification can be made using body color, and body color cannot be used as the most effective morphological character for species identification. In addition, the length of the dorsal papillae, and the bifurcated or single-pointed dorsal appendages were not the most striking and recognizable characteristics of Psychropotes (Hansen, 1975; Xiao, Li & Sha, 2018). In this study, we made a taxonomic key by reviewing morphologically similar features of each species and different features that distinguished them from other congeners. In addition to the stable morphological features of ossicle types, external morphological features were used to identify species, which included the number of dorsal papillae, the presence or absence of free ventrolateral tube feet, the number of ventrolateral tube feet, and the length and position of the dorsal appendages.

The inter- and intraspecific genetic divergences of the COI were calculated to investigate the genetic distances in Psychropotes. For the COI alignment, the intraspecific distances ranged from 0.00 to 1.23%, and the interspecific distances were in the range of 1.38–13.54%. The minimum interspecific distance seemed low, but Bribiesca-Contreras et al. (2022) confirmed that the COI gene seemed to be more conserved in the genus Psychropotes (1.1–13.4%, mean 6.5%), and the interspecific divergence between some species pairs was <2% (e.g., the genetic distances between P. dyscrita and P. moskalevi were 1.38–2.64% in this study, compared to 1.1 ± 0.4% in Bribiesca-Contreras et al. (2022). The genetic distances between P. asperatus sp. nov. and congeners were in the range of 3.14–13.05%, between the new species P. diutiuscauda sp. nov. and congeners in the range of 1.56–12.65%, and between P. nigrimargaria sp. nov. and congeners in the range of 2.84–12.38%. These divergences were higher than the known intraspecific variation in Psychropotes species (0.00–1.23%), which supported the distinction between the three new species and other congeners.

According to the phylogenetic analysis results (Figs. 14, 15 and Figs. S1–S3), although molecular data were unavailable for all species in this group, the existing data indicated that Psychropotes comprises three lineages. Lineage 1: The group of species of the ‘P. longicauda’ morphotype formed a well-supported clade (16S-COI trees: BS = 93, PP = 0.98; COI trees: BS = 100, PP = 1). Within this lineage, two newly identified species were nested: P. diutiuscauda sp. nov., which exhibited a genetic distance of 1.56–4.30% from other species in the group, and P. nigrimargaria sp. nov., with a genetic distance of 2.84–4.85% from other species in the same group. Lineage 2: Another new species, P. asperatus sp. nov., grouped with P. verrucicaudatus. The genetic distance between P. asperatus sp. nov. and the group of species of the ‘P. longicauda’ morphotype ranged from 10.22% to 13.05%, while its genetic distance from the closely related P. verrucicaudatus was 3.14% to 3.26%. Lineage 3 comprised only one species, P. depressa, which consistently formed a separate lineage distinct from the others, its genetic distance from all other species ranged from 10.63% to 13.06%.

The above three lineages divided the species of Psychropotes into three groups, each displaying differences in the range of interspecific genetic distances. Lineage 1 (Group 1): The group of species of the ‘P. longicauda’ morphotype exhibited relatively low COI genetic distances (1.38–6.21%) among its members. Lineage 2 (Group 2): This group, comprising P. asperatus sp. nov., P. verrucicaudatus, and P. cf. semperiana (unavailable COI sequence), showed genetic distances ranging from 10.22% to 13.54% when compared with Group 1. Lineage 3 (Group 3): Represented solely by P. depressa, this group was clearly distinct from both Group 1 and Group 2. Its genetic distances ranged from 11.43% to 13.06% with Group 1 and 9.87% to 11.67% with Group 2. The phylogenetic results aligned with the morphological classifications, thereby strengthening the reliability of our conclusions. Morphologically, each group exhibited distinct characteristics. Species in Group 1 were characterized by long dorsal appendages, positioned either close to or at a short distance from the posterior body. Species in Group 1 were distinguished by the presence of warts on their dorsal skin, while species in Group 3 showed a high degree of morphological variability. However, the limited availability of genetic sequences for deep-sea holothuroids in public databases remains a significant challenge in fully resolving the phylogenetic relationships within Psychropotes. The lack of data for certain species may obscure a comprehensive understanding of their actual genetic divergence.

In addition, the analysis revealed that the family Laetmogonidae is polyphyletic, consistent with the non-monophyletic view proposed by Miller et al. (2017). The results showed poor support, likely due to the limited availability of sequences for both Laetmogonidae and Pelagothuriidae. Phylogenetic trees constructed using a concatenated dataset provided a more accurate representation of interspecific relationships compared to those based solely on the COI gene dataset in this study. Although our study enhanced phylogenetic reliability by integrating two inference methods (ML and BI), further improvements in accuracy require the incorporation of additional molecular markers to mitigate bias inherent in single-marker analyses. Broader sampling and the inclusion of more molecular data are essential for resolving these uncertainties, providing a clearer understanding of the evolutionary history and establishing more distinct species boundaries within this order.

Geographical distribution of psychropotid species

Based on the analysis of available data from OBIS and GBIF, the world distribution of psychropotid species is as follows. In total, there are three genera and 45 species in the family Psychropotidae, which include the three new species described here, 27 of which are distributed in the Pacific Ocean (i.e., 13 species in Psychropotes, 11 species in Benthodytes, three species in Psycheotrephes), 12 species in the Indian Ocean (i.e., nine species in Psychropotes and three species in Benthodytes), seven species in the Atlantic Ocean (i.e., three species in Psychropotes and four species in Benthodytes), and only one species in the Antarctic (i.e., Psycheotrephes recta).

Twenty-three species, which include three new species, in the genus Psychropotes are distributed widely, with a depth range of 957–7,250 m. P. verrucosa has the deepest record in the genus (depth 7,250 m), and P. depressa has the shallowest record (depth 957 m). Thirteen species of Psychropotes were discovered in the deep water of the Pacific Ocean, and five of those species were recorded from the South Pacific Ocean: P. asperatus sp. nov., P. depressa, P. longicauda, P. loveni, and P. verrucosa. Seven species are only distributed in the Indian Ocean: P. belyaevi, P. diutiuscauda sp. nov., P. fuscopurpurea, P. nigrimargaria sp. nov., P. minuta, P. mirabilis, and P. xenochromata. Three other species, P. buglossa, P. scotiae, and P. semperiana, are distributed mainly in the Atlantic and Indian Ocean. Among the four valid species of the genus Psycheotrephes, three were recorded from the Pacific region (Fig. 16), and Psycheotrephes recta (Vaney, 1908) is the only species distributed in the Antarctic (specific data not available). The depth range of Psycheotrephes is 4,182–5,029 m. There are 18 species in the genus Benthodytes, four of which were found only in the Atlantic: Benthodytes gosarsi Gebruk, 2008, B. lingua Perrier R., 1896, B. valdiviae Hansen, 1975, and B. violeta Martinez, Solís-Marín & Penchaszadeh, 2014. Three species are distributed only in the Indian Ocean: B. plana Hansen, 1975, B. superba Koehler & Vaney, 1905, and B. wolffi Rogacheva & Cross in Rogacheva, Cross & Billett, 2009. The remaining species were found mainly in the Pacific Ocean. Benthodytes sibogae Sluiter, 1901a, 1901b and B. marianensis Li et al., 2018 were collected from the shallowest (694 m) and deepest (5,567 m) depths recorded for this genus, respectively, with depth information based on specimens collected in the Mariana Trench and the South China Sea (Li et al., 2018; Xiao, Xiao & Zeng, 2020).

Figure 16 The world distribution of species of Psychropotidae based on OBIS and GBIF data.

(A) Twenty-three species of Psychropotes Théel, 1882. (B) Eighteen species of Benthodytes Théel, 1882. (C) Three species of Psycheotrephes Théel, 1882. Different species are represented by different colors and shapes.

In this study, the number of Psychropotes species identified in the Kermadec Trench has increased to five (i.e., P. verrucosa, P. loveni, P. longicauda, P. depressa, and P. asperatus sp. nov.). In addition, two new Psychropotes species were found in the WZFZ, making the first report of this genus in the region. The description of three new species and one new record provided valuable insights for future studies on species relationships as well as ecological factors such as geological structure, depth, and food chains in the Kermadec Trench and WZFZ, contributing new clues to understanding the ecological patterns in these regions.

Currently, six species (i.e., Psychropotes asperatus sp. nov., P. diutiuscauda sp. nov., P. moskalevi, P. nigrimargaria sp. nov., P. pawsoni, and P. verrucosa) are known to have a vertical distribution extending over more than 6,000 m, spanning the northwest Pacific Ocean, the east Indian Ocean (WZFZ), and the south Pacific Ocean (the Kermadec Trench). Earlier surveys mainly employed sampling methods such as the Agassiz trawl (Agassiz, 1888), beam trawl, detritus sledge (Ockelmann sledge) (Ockelmann, 1964), and epibenthic sledge (Rothlisberg-Pearcy hyperbenthic sledge) (Rothlisberg & Pearcy, 1977; Brandt & Barthel, 1995), which limited the depth range of specimens to some extent. With the development of underwater technology, three main types of submersibles have been used to observe and collect biological specimens: remotely operated vehicles (ROVs), HOVs, and autonomous underwater vehicles (AUVs) (Yang et al., 2022; Zhou & Peng, 2023), allowing for the collection of samples from a broader range of water depths. Overall, species in the family Psychropotidae span a wide range of depths (694–7,250 m), which extend from bathyal depths to the hadal zone. Such high species richness and wide geographical distribution indicate that there are likely more species to be discovered in the Pacific Ocean. Notably, the species diversity of the southwest Pacific, including the Kermadec Trench, has been underestimated before this study. The present study suggested that species of Psychropotes were more widely spread in abyssal regions than previously expected. There is no doubt that it is necessary to conduct more investigations of deep-sea holothuroids in different areas to unravel their taxonomy and phylogeny.

The geographical data in this study were based on taxonomic and ecological records, including specimens identified by professional taxonomists on specimens but not published as formal articles (e.g., P. longicauda, P. depressa), species identified from images (e.g., P. semperiana identified only from imagery by Antonina Kremenetskaia, David L Pawson, Diva J Amon, Amanda F Ziegler), and analysis of molecular data obtained from specimens (e.g., Benthodytes sanguinolenta). Re-examination of these materials is required to enhance the accuracy of the geographical analyses.

Conclusions

We described a new species (Psychropotes asperatus sp. nov.) and one new record (Psychropotes depressa) from the Kermadec Trench region in the South Pacific Ocean, and two new species (Psychropotes diutiuscauda sp. nov. and Psychropotes nigrimargaria sp. nov.) from the WZFZ in the East Indian Ocean. We developed a dichotomous key for Psychropotes. The yellow-green body color, the size of the dorsal papillae, and unpaired dorsal appendages that were single-pointed or divided in varying degrees at the tip could not be used as unique identification features to distinguish the species. The key morphological features were the number of dorsal papillae, the number of free ventrolateral tube feet, the length and position of the unpaired dorsal appendages, and the ossicle types. Phylogenetic analyses showed that the family Laetmogonidae is polyphyletic, and the genus Benthodytes in the family Psychropotidae is paraphyletic. The family Psychropotidae exhibits a broad geographical range, with the Pacific Ocean displaying the greatest variety of species, hosting 27 out of the 44 recognized species.

Supplemental Information

Supplemental Information 1 Primer sequences used for the amplification of 16S and COI in new species and one new record of Psychropotes; F is for the forward and R for the reverse primer.

Supplemental Information 2 GenBank accession numbers, localities, voucher information, and source of all species and sequences used in this study.

Supplemental Information 3 Bayesian inference (BI) phylogenetic tree based on concatenated 16S-COI sequences that shows phylogenetic relationships among elasipodid species.

Numbers near branches indicate the posterior probabilities (PP) of BI. The thin-annotated branches represent the differences in topologies of the ML and BI trees. The branches of different families are represented by different colors. The sequences of three new species and one new record of Psychropotes provided in this study are in red. PP values < 0.5 are not displayed.

Supplemental Information 4 Maximum-likelihood (ML) phylogenetic tree based on COI sequences that shows phylogenetic relationships among elasipodid species.

Numbers near branches indicate the bootstrap values (BS) from ML. The thin-annotated branches represent the differences in topologies of the ML and BI trees. The branches of different families are represented by different colors. The sequences of three new species and one new record of Psychropotes provided in this study are in red. BS values < 50 are not displayed.

Supplemental Information 5 Bayesian inference (BI) phylogenetic tree based on COI sequences that shows phylogenetic relationships among elasipodid species.

Numbers near branches indicate the posterior probabilities (PP) of BI. The thin-annotated branches represent the differences in topologies of the ML and BI trees. The branches of different families are represented by different colors. The sequences of three new species and one new record of Psychropotes provided in this study are in red. PP values < 0.5 are not displayed.

Supplemental Information 6 Raw data: 16S sequences.

Supplemental Information 7 Raw data: COI sequences.

We thank the crew of the vessel ‘Tansuo 1’ and the HOV team ‘Fendouzhe’ for their assistance in the collection of the specimens. We appreciate the scientists from IDSSE and NIWA for their help in taking photographs of fresh specimens before fixation. Many thanks to Prof. Xiaotong Peng group for providing the gold spraying service and Prof. Shenghua Mei group for electron microscope support. We are grateful to Dr. Ashley Rowden, Dr. Daniel Leduc, and Ms. Caroline Chin from NIWA for their constructive comments on the manuscript.

Additional Information and Declarations

Competing Interests

The authors declare that they have no competing interests.

Author Contributions

Yunlu Xiao conceived and designed the experiments, performed the experiments, analyzed the data, prepared figures and/or tables, authored or reviewed drafts of the article, and approved the final draft.

Haibin Zhang conceived and designed the experiments, authored or reviewed drafts of the article, and approved the final draft.

Field Study Permissions

The following information was supplied relating to field study approvals (i.e., approving body and any reference numbers):

The specimens in our study were collected from the Kermadec Trench region and the Wallaby-Zenith Fracture Zone. The Wallaby-Zenith Fracture Zone is located in international waters, where sampling can be conducted without the need for a specific permit. In contrast, sampling in the Kermadec Trench region required appropriate permits. Here is the Kermadec permit information: Specimens were collected under Ministry for Primary Industries Special Permit 842.

Data Availability

The following information was supplied regarding data availability:

The COI sequences are available at GenBank: PP869369 to PP869372. The 16S sequences are available at GenBank: PP868346 to PP868349.

New Species Registration

The following information was supplied regarding the registration of a newly described species:

Publication LSID: urn:lsid:zoobank.org:pub:A7C04F7E-10CB-4FD5-9084-10698440207A

Psychropotes asperatus sp. nov. LSID: urn:lsid:zoobank.org:act:CC3A0572-9A57-407E-95F9-4E84AD9CE68F

Psychropotes diutiuscauda sp. nov. LSID: urn:lsid:zoobank.org:act:DA69574E-6164-4584-A242-B9AB92F66639

Psychropotes nigrimargaria sp. nov. LSID: urn:lsid:zoobank.org:act:738347A2-DD78-41DA-A293-BA5532D06F69.

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
