# Peer review of "Integrative taxonomy reveals three new species and one new record of Psychropotes (Holothuroidea, Elasipodida, Psychropotidae) from the Kermadec Trench region and the Wallaby-Zenith Fracture Zone"

_PeerJ, doi:10.7717/peerj.18806_

## Round 0.1 · original submission · Major Revisions

I enjoyed reading this manuscript. However, I do agree with the reviewers on the issue of additional information and clarity in their morphological descriptions especially of the ossicles as highlighted by 2 of the reviewers. Please clarify and address these comments by Review 1 and 2.

Special care also required for the calculation of genetic distances to cover as many species as possible within the genus albeit there is limitation of data. But please consider the comments of Review 1 and 2.

Do also make correction for all the comments by all reviewers on the formatting and minor mistakes

I am more than happy to accept this manuscript after addressing all these issues.

·

Basic reporting

- Please provide a caption for each figure that is easy to understand without referring to the main text.
- Please check the references. Several paper are lacking, and other some articles are not cited in your main text.

Experimental design

- In calculating genetic distances, I suggest that the number of individuals included should be increased to the maximum extent possible for species with many individuals available, such as P. raripes reported by Gubili et al. (2017) and Gebruk et al. (2020).

Validity of the findings

- In taxonomic description sections, please present morphological measurements in a quantitative description, such as by stating the number of values and the average.

Additional comments

This study is very useful that contributes to understanding the diversity of deep-sea benthos by providing a morphological description of Psychropotes from the Kermadec Trench area and the eastern Indian Ocean, where knowledge has been scarce to date, and summarizing the distribution records of three genera of Psychropotidae for the first time in about half a century since Hansen (1975). However, there are some issues that need to be improved in the current manuscript, and these need to be improved in order to promote scientific discussion in this research field. Taxonomic research on Psychropotes had been in a stagnant state for a long time after Hansen (1975), but in the past decade, important molecular phylogenetic and morphological studies have been published one after another. However, the classification of this genus has been discussed based on the morphological descriptions of a small number of individuals, and some morphological differences between species are ambiguous. In order to advance objective discussion in the future, it is essential to introduce quantitative descriptions like those seen in other holothurians.

I have a few general comments, and several minor and specific comments are added on an attached file.

1. Please reconsider any information that is essential to include in the Introduction. In the paragraphs describing the taxonomic information of psychropotid sea cucumbers in Kermadec Trench and WZFZ, abyssal and hadal studies essential to this study, such as Hansen (1975), are missing, while many intertidal studies that are less relevant to this study are cited.
2. In taxonomic description sections, please present morphological measurements in a quantitative description, such as by stating the number of values and the average. For specimens that clearly show variation within individuals, such as ossicle morphologies, please provide a frequency distribution so that it is possible to determine whether the observed characteristics are frequent or rare within the observed tissue/specimen. All three new species described in this study are described based on the holotype only, and it is difficult to determine whether your observed characteristics are representative of the species based on the current description.
3. In calculating genetic distances, I suggest that the number of individuals included should be increased to the maximum extent possible for species with many individuals available, such as P. raripes reported by Gubili et al. (2017) and Gebruk et al. (2020). At this time, the number of individuals discussed is extremely limited, and the calculations appear to have been performed using an insufficient data set to determine the inter/intra-specific genetic range of Psychropotes. The validity of the three newly described species should be discussed based on the recalculated inter/intra-specific genetic range in present genus.
4. In discussing the distribution of known species of Psychropotes, please briefly explain how the addition of the four distribution records in your study has improved our understanding of the distribution and species diversity of this genus. For example, the species diversity of the SW Pacific, including the Kermadec Trench, has been underestimated before your study.
5. Please provide a caption for each figure that is easy to understand without referring to the main text. The SEM images of ossicles in the description of P. aperatus sp. nov. lack explanations.
6. Please check the references. Several paper are lacking, and other some articles are not cited in your main text.

Reviewer 2 ·

Basic reporting

Manuscript of Xiao and Zhang is an important contribution to our knowledge of Psychropotes, a taxonomically difficult holothuroid deep-sea genus. It includes very detailed morphological descriptions, well illustrated , and also contains new data on two mitochondrial markers and the phylogenetic analysis. I think that the manuscript can be published in PeerJ after minor revision. My comments are mostly technical and do not affect the overall positive impression obtained from reading of the manuscript.

Comments to the Authors:

1) The genus Psychropotes includes the "longicauda"group of species with a tail located on very posterior dorsum. Apart from P. longicauda the group comprises P. raripes, P. fuscopurpurea, P. buglossa, P. moskalevi, P. dyscrita, P. xenochromata, P. pawsoni, P. dubiosa, and also P. diutiuscauda and P. nigrimargaria. Low COI genetic distances are known only for this species group, not for all species of the genus. Any notices on presence of this species group, its phylogeny and affinity to the new species are completely missing in the manuscript. According to your phylogenetic analysis results, the 'longicauda' group formed a well-supported clade (Fig. 15 in the manuscript)


2) Psychropotes buglossa E. Perrier, 1886 was recently re-described. It can be added to the key. See:
(SOSA) SOSA et al. (2024)
Ocean Species Discoveries 1–12 — A primer for accelerating marine invertebrate taxonomy. Biodiversity Data
Journal 12: e128431. https://doi.org/10.3897/BDJ.12.e128431


3) Some of Psychropotes species have two ossicle layers in the body wall skin. Ossicles from deeper skin layers are usually more gracile and flat. Some ossicles illustrated in the manuscript can be from deeper skin layers, for example fig.6 A-C. Please provide additional information how ossicles are arranged in the skin.

Experimental design

no comment

Validity of the findings

no comment

Additional comments

-References - please use alphabetical order

-Please also keep the “–“ and “-“ dashes consistent and used correctly throughout the work



lines 18-19 "was the dominant group in the deep-sea benthic fauna at lower bathyal-abyssal depths throughout the global oceans"

Dominant group usually means that they dominate either in biomass or in number. Although few such cases have been described, dominance of this group is not a common phenomenon. But they can be prominent representatives at the abyssal depths.


lines 64-65 "This genus has been found at hadal depths (6135-7250 m) across all non-polar oceans"
please specify, which species have been found at hadal across all the non-polar oceans. As I know, except new species described in the manuscript, only two species are known from the hadal: P. moskalevi and P. pawsoni (both from the Pacific only).


line 65 and throughout the text (except species authorities) Gebruk et al., 2020

line 92 or 105 please provide the name of the research vessel and cruise number

lines 109-111 Please provide Cat. Nrs for specimens deposited at IDSSE

line 143 please provide extended name for the genetic markers


line 151 did you compute the models for the unpartitioned COI alignment or separately for the positions 1-3?


line 203 Brim is composed by fused tube feet. Please rephrase

line 258 Please provide storage place and Cat. number for the holotype

line 319 P. xenochromata

Line 325 is question mark needed?

lines 393-409 I suggest to give the references in the Synonymy part in a plain text, not italics . According to ICZN: "The genus and specific name are conventionally written in italics (or other contrasting typeface) to distinguish the name from surrounding text."
At the moment all text here is italicised and taxon names are not contrasting

line 436 Hansen, 1975 [insert comma after Author name]

lines 436 and 452 Xiao et al., 2018

line 732 Elpidia atakama - in italics
line 752 Elpidiidae (capital letter)
line 832 Shirshov PP is not authored this reference
line 883 please remove duplicate ref.

·

Basic reporting

No comment

Experimental design

No comment

Validity of the findings

No comment

Additional comments

-Line 18. “The holothuroid genus Psychropotes is the largest genus in the family Psychropotidae. Prior to this study, this family contained 19 accepted species” This is a mistake, there are many more species within the family. The authors probably are talking about the genus Psychropotes. See line 47 (The family Psychropotidae includes 41 accepted species), then see line 618.

-Lines 393 to 409, review with the Journal where to use italics in the text.

-Line 432. Distribution. Add references to the geographic and bathymetric information.
-Line 656. It says "We developed an dichotomous key", it should say "We developed a dichotomous key".

---

## Round 0.2 · accepted · Accept

I am satisfied with all the improvements and revisions done for your manuscript. This manuscript is well-written and all comments were addressed.